# Sudden Infant Death Associated with Rhinovirus Infection

**DOI:** 10.3390/v16040518

**Published:** 2024-03-27

**Authors:** Christelle Auvray, Stéphanie Perez-Martin, Isabelle Schuffenecker, Cécile Pitoiset, Georges Tarris, Katia Ambert-Balay, Laurent Martin, Nathalie Dullier-Taillefumier, Jean-Baptiste Bour, Catherine Manoha

**Affiliations:** 1Department of Microbiology, Virology Laboratory, Dijon University Hospital, 21070 Dijon, France; christelle.auvray@chu-dijon.fr (C.A.); cecile.pitoiset@chu-dijon.fr (C.P.);; 2Department of Pediatrics, Dijon University Hospital, 21079 Dijon, France; stephanie.perez-martin@chu-dijon.fr; 3French National Enterovirus/Parechovirus Reference Centre, Hospices Civils de Lyon, 69317 Lyon, France; isabelle.schuffenecker@chu-lyon.fr; 4Department of Pathology, Dijon University Hospital, 21070 Dijon, France; georges.tarris@u-bourgogne.fr (G.T.); laurent.martin@chu-dijon.fr (L.M.); nathalie.taillefumier@chu-dijon.fr (N.D.-T.); 5French National Reference Centre for Gastroenteritis Viruses, Virology Laboratory, University Hospital of Dijon, 21070 Dijon, France; katia.balay@chu-dijon.fr

**Keywords:** infant, sudden death, rhinovirus, cerebrospinal fluid, fatal, virus, central nervous system, infection, SIDS

## Abstract

A less than one-month-old infant with symptoms of rhinitis died unexpectedly in his sleep. He was not born prematurely and had no known underlying disease. Cerebrospinal fluid, nasopharyngeal and lung samples, and rectal swab were found to be positive for subgroup A rhinovirus, while the blood was negative. This case highlights the important finding that the rhinovirus, a common pathogen associated with upper respiratory tract infections, can sometimes, as the only pathogen, lead to complications such as a cerebrospinal infection and be involved in the sudden infant death syndrome (SIDS). Vigilance is necessary in case of viral infections in the infant’s environment, and measures of hygiene and protection must be encouraged in order to reduce the risk of the SIDS.

## 1. Introduction

Rhinoviruses (RVs) are the most common cause of the common cold in children and adults. The most common clinical characteristics in patients with RVs include rhinorrhea, nasal congestion, cough, sore throat, headache, and acute otitis media episodes. However, RVs may cause more serious manifestations including bronchiolitis, bronchitis, exacerbations of asthma, wheezing, and pneumonia [1]. Moreover, studies show that RVs are an important pathogenic factor for the development of recurrent wheeze and subsequent asthma [1]. RVs play a predominant role as respiratory pathogens in early life. Asymptomatic RV infections occur in healthy infants without nasal symptoms, and though RVs can be found in up to 20% of infants [2], their impact on morbidity is unknown. In infants and toddlers, RVs are the pathogens most commonly associated with upper- and lower-respiratory tract infections and wheezing in the first year of life [3]. The highest rate of hospitalization among children with RV was reported for infants aged 0–5 months [4], and host factors associated with more severe RV-associated illness in infants included maternal and family history of atopy [5]. Allergy and asthma are major risk factors for more severe RVS-related illnesses, as well as a coinfection with respiratory syncytial virus (RSV), in children under two years [6,7]. Rarely, encephalopathy cases linked to rhinovirus have been reported consistently in children [8,9,10,11]. 

The sudden infant death syndrome (SIDS) is defined as a sudden death of an apparently healthy infant under the age of one year. The fact that death in the first year is 20 times more frequent than death from any cause between the ages of 1–18 years highlights the importance of identifying features and risks of the SIDS. The RV-related sudden death is exceptionally rare. We report here the case of an infant with an RV respiratory infection, which spread to the cerebrospinal fluid (CSF) and was very likely a trigger of the sudden death. 

## 2. Case Report

The patient was a 20-day-old male infant who was born near full-term following normal development. He had no family history of asthma or atopic status. The mother was 16 years old. He was exposed to cigarette smoke at home. The day before death, the infant was taken to see a physician. The general practitioner reported rhinitis and congestion. On the day that the death occurred, the infant initially appeared well and was put to sleep in a safe position. The infant was found cold and white in his bed, in the supine position. Emergency services were called. On site, the infant was intubated; gamma adrenalin was administered, and the infant was in asystole. Following the sudden cardiopulmonary arrest that occurred at home, he was then taken to hospital.

Before the autopsy, which was performed the day after death, a whole-body CT scan revealed no lesion or fracture. Areas of frosted glass and condensation were distributed over both lungs that could be due to an infection or post-mortem modifications. The whole-body scanner did not reveal encephalitis. Autopsy included external examination and the removal of body organs for microbiological and histopathological analyses. Extrinsic causes of death such as child abuse or trauma were ruled out. Post-mortem samples were collected according to the current recommended guidelines. Pre-autopsy samples taken by physicians that received the infant at the emergency department were from the upper (nasopharyngeal swab (NPS), nasal secretions (NS), throat) and lower respiratory tracts (bronchoalveolar lavage (BAL)), digestive tract (rectal swab), cerebrospinal fluid (CSF), blood, muscle, and heart (sampling was performed with a trocar). Samples were analyzed upon reception at the laboratory. To reduce the viscosity of BAL, sterile PBS was added. A 200 μL volume of sample was analyzed. The blood was diluted to one-tenth in PBS, and a 200 μL volume was extracted. One ml of DPBS was added to the rectal swab, the sample vortexed, and then centrifuged for 15 min at 10,000× *g* at room temperature. An 800 μL volume of supernatant was collected for extraction. Samples taken by the pathologist during autopsy (at-autopsy samples) were from the lower respiratory tract (lung), cardiac system (heart), abdominal organs (liver), and urinary tract (kidney). The collected biopsies were kept frozen before analysis. After thawing, they were ground with a pestle, vortexed, treated for 10 min at 56 °C with 50 μL of proteinase K treatment, and then centrifuged for 1 min at 10,000 rpm. The obtained supernatant was analyzed. 

Among blood parameters, the levels of C-reactive protein (CRP) were in the normal range (Table 1). Red blood cell, hemoglobin, and platelet counts were normal. Bacterial cultures of blood specimens yielded negative results. Hematological analyses excluded hematological malignancy. IgE specific to the detection of cow’s milk was negative. The percentage of T lymphocytes was 49%, and the ratio of CD4/CD8 was 2.2. The phenotypic analysis by flow cytometry found no suspicious lymphocyte population. Biochemical analyses in blood revealed no disturbance except elevated phosphorus (252 mg/mL [57–79]) and potassium (>10 mM [3.5–5.2]). The cell count was not performed in the CSF, which was slightly bloody; it was centrifuged before further analyses could be performed. In the clear CSF, proteins were elevated (1.21 g/L [0.12–0.6]). The high level of lactate (1964 mg/L [54–198]) and the low level of glucose (0.17 g/L [0.4–0.7]) could suggest bacterial meningitis, tissue hypoxia, seizures, or hemorrhage. 

Microbiological investigations were performed on samples collected at death and during the autopsy. None of the microbiological cultures and investigations yielded any pathogenic bacteria in the patient’s blood, CSF, urine, rectal swab, throat, tracheal aspiration, or lung biopsy specimens. Cultures from lung biopsy specimens were negative for mycobacteria. 

The samples from the liver, heart, lung, BAL, rectal swab, kidney, and serum were tested for enterovirus (EV) by RT-PCR using RIDAgene (R-Biopharm AG^®^, Darmstadt, GER), and all samples were negative. In the CSF, EVs were detected using the BioFire^®^ film array meningitis/encephalitis panel (bioMérieux, Salt Lake City, UT 84108, USA); all the other pathogens were negative. The tests for EVs were performed by the French National Reference Centre for gastroenteritis viruses. Tests for the presence of norovirus, rotavirus A, sapovirus, human astrovirus, and bocavirus were performed using specific semi-quantitative in-house RT-qPCR assays (NF EN ISO 15189 certified), as previously performed (Appendix A), while adenoviruses and parechovirus were tested using the Adenovirus R-gene and Parechovirus R-gene PCR kits (bioMérieux^®^). Internal quality controls (IQC) were used to validate the assays. All enteric RT-PCRs were performed on a QuantStudio^©^ 5 Real-Time PCR System (Applied Biosystems, ThermoFisher Scientific^®^, Waltham, MA, USA). Gastroenteritis viruses and SARS-CoV-2 were not found in rectal swabs (Table 2). Cycle threshold (Ct) was used as an inverse proxy for viral load, with lower Ct correlating with higher viral load. Only the rhinovirus test was positive in the nasopharyngeal swab sample using the Panther instrument (Hologic^®^, Marlborough, MA 01752, USA) and a Ct value of 16 was found that suggests a high viral load, while all tests for other respiratory viruses were negative (Table 2). Nasal secretions were tested using the multiplex PCR assay BioFire^®^ Film Array Respiratory Panel 2.1-plus; RV and EV were the only pathogens that were detected in the secretions. To exclude a potential contamination of the CSF by blood, we performed a PCR using Rhino&EV/Cc R-gene^®^ kit (Biomérieux), and the rhinovirus genome was not detected. Blood was RV negative. To exclude false-negative results, blood was spiked with an RV-positive sample to detect potential inhibitors. The Ct obtained for the known RV-positive sample and for the known RV-positive sample spiked into the infant blood were similar. Moreover, the Ct obtained for cellular control in the blood and for cellular control in the blood spiked with the RV-positive sample were equal. 

For genotyping, nucleic acids were extracted from nasopharyngeal swab samples, rectal swabs, and CSF specimens with the NucliSENS easyMAG automated system according to the manufacturer’s instructions (BioMérieux, Lyon, France). A nested PCR assay specific for the RV VP4/VP2 genomic region [12], performed by the French National Reference Centre for Enteroviruses, was used for RV typing in the nasopharyngeal samples, rectal swabs, and CSF specimens. 

Human RV type A 10 (RV-A10) strain was identified in the three samples by sequencing, and phylogenetic analysis confirmed the genotype assignment. The amplification of the genomic regions encoding the complete gene sequences of EV-B and the partial gene sequences of EV species (EV-A to EV-D) associated with human infections [13,14] was negative. Genotyping results excluded an EV infection, allowing us to conclude regarding the detection of RV-A10 rather than EVs by the film array meningitis/encephalitis panel. It should be mentioned that some cross-reactional detection of RVs and EVs can occur (conserved regions are often targeted in PCR assays and found in picornaviruses, EVs, and RVs), leading to the substantial cross-amplification of EVs by RVS primers and vice versa.

In the CSF, the detection of the rhinovirus was thus confirmed by two different RT-PCR methods: film array (multiplex real-time PCR) and an end-point PCR for typing and sequencing.

These extensive microbiological investigations failed to reveal the presence of any bacteria or virus other than RV following the death of the infant from the SIDS (Table 2).

The gross analysis of the central nervous system (CNS) was normal. The gross examination of the lungs revealed diffuse parenchymal congestion. Histopathology revealed histiocytic alveolitis associated with signs of acute hypoxia. RVs were detected inside macrophages (Figure 1). 

We then performed a PCR testing using Rhino&EV/Cc R-gene^®^ kit (bioMérieux) on BAL fluid, in which the rhinovirus genome was found that confirms the rhinovirus infection in the lower airways. Using R-gene assays (bioMérieux), in the BAL, despite de facto strong dilution, a Ct value of 31 was found, and in the nasal secretions, which are also diluted because of viscosity, a Ct of 25 was obtained that suggests a high viral load in both the upper and lower airways. Blood was RV negative. 

## 3. Discussion

This is the first report of an SIDS case associated with the presence of the RV in respiratory, CSF, and enteric samples, identified after a thorough investigation of pathogens. Indeed, we searched for a large range of pathogens using sensitive real-time PCR assays that were able to exclude coinfection, underlining the role of the rhinovirus in the SIDS. 

In the present case, the infant was in the late neonatal period at the time of death. In the global neonatal mortality estimates, preterm birth and intrapartum complications accounted for the majority of deaths in the early period (0–6 days), while in the late neonatal period (7–27 days), nearly half of all deaths occurred from infectious causes [15]. 

Before the year 2000, the SIDS cases associated with RVs in respiratory samples were infrequently described. One case was an 11-month-old infant with symptoms of mild asthma with RV-A47 in respiratory samples who died unexpectedly during sleep [16]. Two similar infant cases were reported by Urquhart [17]. Another sudden case associated with one positive RV sample was reported in a 5-month-old infant who presented respiratory distress [18]. It is worth noting that viruses are not frequently detected in the CSF when sudden death occurs; in a recent review focused on children younger than 2 years in France, only 2% of the CSF samples tested positive for viruses [19]. 

While the RV-positive CSF samples have not been previously reported in the context of the SIDS, the RV-positive CSF was found in two boys under 2 years of age with encephalitis (one also had streptococcal pharyngitis) and in a 5-month-old girl that presented with meningitis and in whom a coinfection with Haemophilus influenzae was detected [9,20,21]. All these cases, including ours, are in favor of the potential involvement of the CNS in an RV infection. It was recently shown that in vitro microglia, the main resident immune cells in the CNS, were susceptible to infection with an RV-A strain, which also suggests the potential for RV infection in the CNS [22].

A probable CNS involvement has also been suggested when the RV was the only detected pathogen in respiratory samples. Two infants aged 3 months with encephalitis had RV strains A9 and B52, both identified in nasopharyngeal swabs and, for one, in a stool sample. RV was thought to be responsible for encephalitis [23]. Previously, three young patients aged 2 to 29 months with meningoencephalitis were found to have RV-1B and 2 in throat swabs [10]. In a two-year-old girl with mild encephalitis/encephalopathy and a reversible splenial lesion (MERS), RV-A49 was identified in the serum, while no respiratory signs were visible [11]. The other cases involved older children [8,24]. All these cases are in favor of the potential CNS involvement in RV infections. 

In our report, RV was detected in upper respiratory samples, rectal swabs, and the CSF. While the virus is frequently found in feces [25], it is rarely detected in the CSF. The means by which the RV reaches the CNS is unknown. The olfactory nerve connects the nasal cavity directly with the CNS, and thus, RVs may enter via the olfactory route, which is used notably by influenza and SARS-CoV-2 [26,27]. The virus may also go through the hematogenous blood–brain barrier (BBB). Rhinoviremia has been suspected to favor its dissemination to the brain [24,28]. In this case, we did not detect RV in the blood either with an EV PCR (RIDA-gene) or an RV/EV PCR (Rhino&EV/Cc R-gene); however the viremia may have been at a level below the sensitivity of these assays, or fleeting, or perhaps there was truly no viremia. 

Then, we tried to define the risk factors that may have contributed to the SIDS in this case. There are three types of risk factors for the SIDS. The first group of risk factors is linked to the critical development period with the establishment of the central nervous system’s homoeostatic mechanisms that control arousal and the cardiorespiratory function [29,30]. The second group encompasses the child-related factors, such as male gender, prematurity, genetic polymorphism, and young maternal age. The third group is composed of extrinsic factors, and the most common factors are prone positioning of a child, smoking exposure during pregnancy, use of soft mattresses, bed sharing, elevated ambient temperature, and viral infections. Viral infections can induce different mechanisms, potentially leading to sudden death, the chemokine-induced enhancement of BBB permeability, diurnal variations in the neuroendocrine pathway, vascular tone abnormalities, impaired immune maturation, imbalanced immune response, and systemic inflammatory response syndrome [31]. Cytokines and inflammatory chemokines (IL-1ra, IL-6, TNF-α, IL-8, G-CSF, MCP-1/CCL2, and PDGF) have been shown to be elevated in the CSF samples of pediatric patients in whom viral infection led to the SIDS in comparison with the control groups [32]. Unfortunately, we could not test for cytokines and chemokines in the CSF. Moreover, RV infection in neonates was shown to be an important risk factor for apnea [33]. Here, alveolitis and signs of acute hypoxia were observed. A dysregulated immune response toward rhinovirus infection, with excessive inflammation and damage to the epithelial cells, may have resulted in immunopathology and impaired gas exchange in the infant, leading to its death.

Sleep position, especially prone sleep position, places an additional physiological stress on cardiorespiratory systems. In the case we described here, the baby was put to sleep in a safe supine position and was found in this position. However, he had several risk factors: male gender, the intrinsic vulnerable development period, his mother’s young age, a viral infection, and exposure to cigarette smoke. It has been shown that male infants are at a greater risk of developing the SIDS than female infants, along with the infants of young mothers (especially with a maternal age of <20 years) [34,35]. Maternal smoking during pregnancy and passive post-natal exposure to cigarette smoke are significantly associated with the SIDS [36]. Moreover, exposure to cigarette smoke enhances susceptibility to respiratory virus infections [37]. 

We report here a fatal case of SIDS associated with human rhinovirus A. This is the first report of a sudden death in an infant with a rhinovirus-positive cerebrospinal fluid. While the cause of death was certainly multifactorial, RV was likely a contributor to this fatal outcome.

## 4. Conclusions

This case underlines that RV infection may be a risk factor for the development of the SIDS. Some key measures could help to avoid perinatal viral infection and to potentially reduce the risk of the SIDS. These include the reduction in exposure to cigarette smoke and the monitoring of the infant’s environment in order to implement reinforced measures of hygiene and protection in case of viral infections.

## Figures and Tables

**Figure 1 viruses-16-00518-f001:**
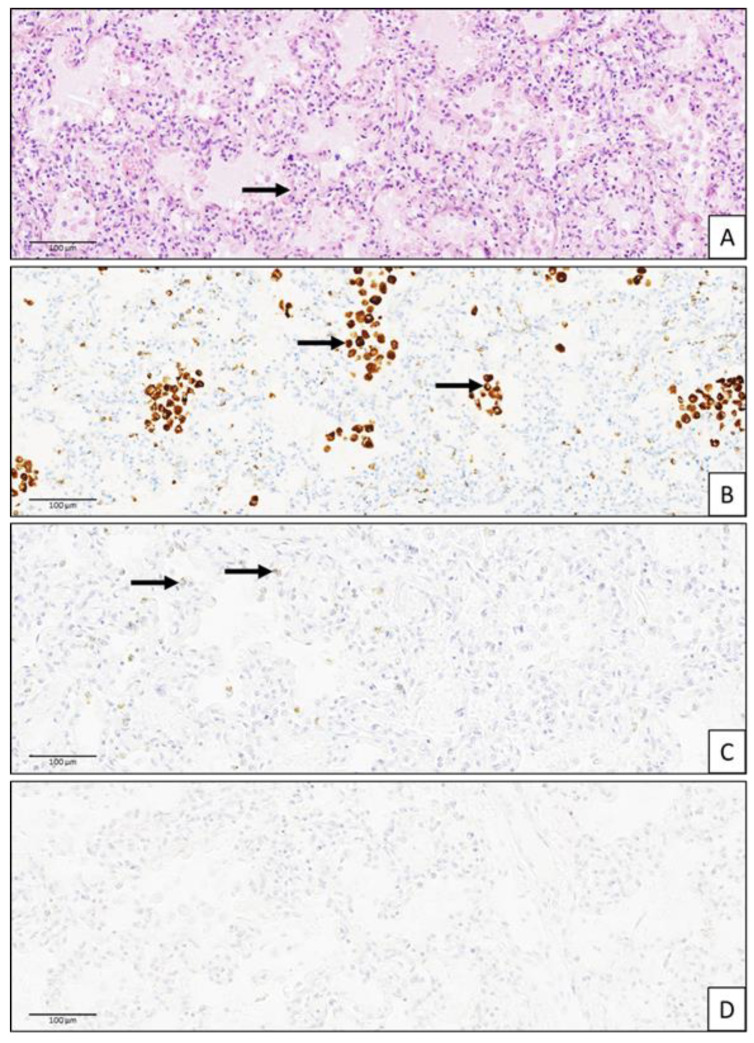
Histopathology of an autopsy lung sample of a 20-day-old boy in the context of sudden death linked to a rhinovirus infection. (**A**) HES, X150: histopathology showed pulmonary diffuse alveolar damage with a hyaline proteinaceous exudate in the alveoli and alveolar membranes, associated with luminal infiltration by numerous macrophages (arrows). (**B**) Anti-CD68, X1500: immunohistochemistry with anti-CD68: IgG1, clone KP1, and reference GA609 (DAKO) revealed numerous CD68+ macrophages inside alveoli (arrows). (**C**) Anti-rhinovirus antibody, X150: the rhinovirus was focally detected inside macrophages using anti-RV antibody: IgG1, clone 4966, and reference ABIN5688681 (Antibodies-online), targeting the VP3 viral protein (arrows). (**D**) Negative stain, X150: specific primary Ab was replaced by PBS and used as the negative control. The scale bar was 100 μm.

**Table 1 viruses-16-00518-t001:** Blood parameters.

Blood Parameters (Day of Sudden Death)		Normal Range
WBC (×10^3^/μL)	8.6	5.0–20.0
RBC (×10^12^/L)	3.5	3.5–6.1
Hemoglobin (g/100 mL)	12	12.0–20.5
Platelets (×10^3^/μL)	210	150–450
NEU (×10^3^/μL); %	1.3; 15.2%	[1.0–9.0]
LYM (×10^3^/μL); %	6.51; 75.8%	[2.2–16.8]
MONO (×10^3^/μL); %	0.65; 7.6%	[0.05–1.1]
EOS (×10^3^/μL); %	0.06; 0.8%	[0.0–0.85]
CRP (mg/mL)	<4	< 4

WBC, white blood cell; RBC, red blood cell; NEU, neutrophil; LYM, lymphocyte; MONO, monocyte; EOS, eosinophil; and CRP, C-reactive protein.

**Table 2 viruses-16-00518-t002:** Molecular assays performed for the identification of pathogens.

	Pre-Autopsy Samples									At-Autopsy Samples				
	Throat	BAL	NS	NP Swab	Rectal Swab	Muscle	Heart	Blood	Serum	CSF	Liver	Heart	Lung	Kidney
Film array meningitis/encephalitis panel * (Biofire, bioMérieux)										EV				
EV (RIDA®gene, R- Biopharm)														
Bocavirus, Rotavirus A, Aichivirus														
Norovirus, Sapovirus, Astrovirus														
Parechovirus (R-gene, bioMérieux)														
CMV (R-gene, BioMérieux)														
HSV-1, HSV-2 (R-gene, bioMérieux)														
EBV (R-gene, BioMérieux)														
Parvovirus B19 (R-gene, BioMérieux)														
HHV-6 (R-gene, BioMérieux)														
Influenza A and B (Panther, Hologic)														
RSV, hMPV (Panther, Hologic)														
Parainfluenza 1, 2 3, and 4 (Panther, Hologic)														
RV/EV (Panther, Hologic)				RV										
SARS-CoV2 (Panther, Hologic)														
AdV (R-gene, BioMérieux)														
Rubella IgM (VIDAS® BioMérieux)														
*M. pneumoniae* IgM (Virclia)														
Film array respiratory panel RP2 plus ** (Biofire, bioMérieux)			RV											
RV/EV Typing and sequencing				RV	RV					RV				
RV&EV/Ctrl cell (R-gene, BioMérieux)		RV	RV					RV-neg						

In gray: assay used, negative result; in red: assay used, detected pathogen; in white: no assay. * The film array meningitis/encephalitis panel (Biofire) allows the detection of bacteria such as Haemophilus influenza; Streptococcus pneumonia; Neisseria meningitides; Streptococcus agalactiae; Escherichia coli; and Listeria monocytogenes; viruses such as EV; HSV-1/2; VZV; HHV-6; HPeV; and CMV; and a fungus, i.e., Cryptococcus neoformans/gattii. ** The Film Array Respiratory Panel RP2 plus (Biofire) allows the detection of viruses such as AdV, hMPV, RSV, RV/EV, Influenza A (H1, H1-2009), Influenza B, Parainfluenza 1 to 4, human coronavirus NL63, 229E, OC43, HKU1, SARS-CoV-2, and MERS-CoV and bacteria such as Bordetella pertussis, Bordetella parapertussis, Chlamydia pneumoniae, and Mycoplasma pneumonia.

## Data Availability

No new data were created or analyzed in this study. Data sharing is not applicable to this article.

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
