# Peer review of "Sudden Infant Death Associated with Rhinovirus Infection"

_viruses, 2024, doi:10.3390/v16040518_

Round 1
Reviewer 1 Report
Comments and Suggestions for Authors
The presented case report of SID caused by Rhinovirus infection is truly interesting and contributes significantly to the field. The Authors clearly introduce the case, giving very good information about the patient as well as provide the reader with good and merit full discussion. After going over the whole draft, I have some minor comments that should be addressed within the revision.
1) Please provide scale for Figure 1 as well as please describe what arrows mean in the figure legend.
2) Line 106 doesn't need to be bolded.
3) Line 78 - what does it mean that CRP was normal? it should say the levels of CRP were normal/in normal range.
4) Line 227 - how less than 1-month old infant could give a formal consent for the participation in the study?
5) The reference list is very long as for case report paper. Please consider shortening and keeping only key references.
Comments on the Quality of English LanguageSome minor grammatical errors are present within the draft. Please review them carefully.
Author Response
1) Please provide scale for Figure 1 as well as please describe what arrows mean in the figure legend.
As requested, a scale has been added to Figure 1 and legend has been modified.
2) Line 106 doesn't need to be bolded.
Done
3) Line 78 - what does it mean that CRP was normal? it should say the levels of CRP were normal/in normal range.
The correction has been made.
4) Line 227 - how less than 1-month old infant could give a formal consent for the participation in the study?
The reviewer is right, it is now indicated correctly as follows:
The parents were informed and consented to the participation in this study.
5) The reference list is very long as for case report paper. Please consider shortening and keeping only key references.
As suggested we removed some references
Reviewer 2 Report
Comments and Suggestions for Authors
The case report from Christelle Auvray et al. summarized a description of a Sudden Infant Death Syndrome that affected a 20-day-old boy who was received at the pediatric intensive care unit due to the occurrence of sudden cardio-pulmonary arrest at home that progressed to the patient’s death. The report brings a group of isolated indications that the patient has superior and inferior airways affected by rhinovirus, concomitant with the appearance of rhinovirus genetic material in cerebrospinal fluid. Additionally, the authors discuss the possible involvement of the olfactive nerve in the transposition of the superior airway rhinovirus infection to cerebrospinal fluid and consequently to the central nervous system.
The result showing rhinovirus being detected in a histologic section of the lungs could not be confirmed by any other method present, the same applies for the detection of the rhinovirus genome in the cerebrospinal fluid, which was not confirmed by any other means. ELISA or other orthogonal method should be employed to confirm or not the presence of the virus analyzed.
Additionally, the text is somehow confusing due to a lack of punctuation and incorrect use of the word “detected”, making it unclear if the authors are referring solely to the methodology used for the detection of a virus or if they are referring to a positive detection of the prementioned virus.
Based on the above, I don`t recommend the acceptance of the manuscript in its current form for publication on Viruses.
Major issues:
Diverse statements in the introductory section must be supported by proper references.
The description of the patient´s symptomatology and the more detailed description of the events that preceded the ICU admission could be present as valuable information for education purposes.
The methodology used in the totality of the presented results is not properly described impacting the reproducibility of the analysis:
e.g., i) absence of a detailed description of the antibodies used for the immunolabeling in the histopathologic analysis. ii) inexistence of information about the treatment of collected samples used for virus detection. iii) the use of the terminology pre-autopsy and at-autopsy is very confusing. iv) inexistence of information on the primers used for any of the viruses investigated.
The micrograph panels should present a scale bar.
Results of the PCR-based analysis showing the values obtained (e.g., viral gene copies) by the different equipment/techniques should be disclosed in the text body or a supplementary data set.
Minor issues:
According ICTV and PSG resolution in 2012 the nomenclature of human rhinovirus (HRV) was changed to rhinovirus (RV) (https://www.picornastudygroup.com/proposals/2012/2011.018a,bV.U.v2.Enterovirus-Sp,Ren.pdf / https://ictv.global/report/chapter/picornaviridae/picornaviridae/enterovirus).
Acronyms must be fully spelt the first time they appear in the text.
Comments on the Quality of English LanguageThere are some places in the manuscript with awkward phrasing. The authors should have a native English speaker edit the manuscript prior to its re-submission.
Author Response
The case report from Christelle Auvray et al. summarized a description of a Sudden Infant Death Syndrome that affected a 20-day-old boy who was received at the pediatric intensive care unit due to the occurrence of sudden cardio-pulmonary arrest at home that progressed to the patient’s death. The report brings a group of isolated indications that the patient has superior and inferior airways affected by rhinovirus, concomitant with the appearance of rhinovirus genetic material in cerebrospinal fluid. Additionally, the authors discuss the possible involvement of the olfactive nerve in the transposition of the superior airway rhinovirus infection to cerebrospinal fluid and consequently to the central nervous system.
As described by the reviewer, the patient has superior and inferior airways affected by rhinovirus, concomitant with the appearance of rhinovirus genetic material in cerebrospinal fluid; rhinovirus was also detected in feces
We discussed two possibilities for the transposition of the rhinovirus to the central nervous system, via the olfactive nerve or via viremia
The result showing rhinovirus being detected in a histologic section of the lungs could not be confirmed by any other method present, the same applies for the detection of the rhinovirus genome in the cerebrospinal fluid, which was not confirmed by any other means. ELISA or other orthogonal method should be employed to confirm or not the presence of the virus analyzed.
The rhinovirus was detected in a histologic section of the lungs. To confirm rhinovirus infection in the lower airways, we performed a PCR testing using Rhino&EV/Cc r-gene® kit (Biomérieux) on BAL fluid, a strongly diluted sample in which the rhinovirus genome was found at a Ct=31.
We added a comment in the body text
In the CSF, the detection of the rhinovirus was confirmed by two different RT-PCR methods: film array (multiplex Real time PCR) and an end-point PCR for typing and sequencing. None of these techniques provides a quantitative result.
Additionally, the text is somehow confusing due to a lack of punctuation and incorrect use of the word “detected”, making it unclear if the authors are referring solely to the methodology used for the detection of a virus or if they are referring to a positive detection of the prementioned virus.
The text has been reviewed and corrected to avoid confusion.
Major issues:
Diverse statements in the introductory section must be supported by proper references.
Some references were removed.
The description of the patient´s symptomatology and the more detailed description of the events that preceded the ICU admission could be present as valuable information for education purposes.
The day before death, the infant was taken to see physician for congestion. The next day, the infant was found cold and white in his bed, in the supine position. Emergency services were called. On site, the infant was intubated; Gamma adrenalin was administered, the infant was in asystole. He died and was then taken to hospital.
The methodology used in the totality of the presented results is not properly described impacting the reproducibility of the analysis:
e.g.,
- i) absence of a detailed description of the antibodies used for the immunolabeling in the histopathologic analysis.
As requested, we added information on antibodies to figure legend
Anti-CD68: IgG1, Clone KP1, reference GA609 (DAKO)
Anti-RV: IgG1, Clone 4966, reference ABIN5688681 (Antibodies-online)
- ii) inexistence of information about the treatment of collected samples used for virus detection.
NS, BAL and CSF were analyzed upon reception at the laboratory. To reduce the viscosity of BAL, sterile PBS was added and the sample vortexed. 200 µl of sample was analyzed
The collected biopsies were kept frozen before analysis. After thawing, they were ground with a pestle, vortexed, treated 10 min at 56°C with 50µL of proteinase K treatment, then centrifuged for 1 min at 10,000 rpm. The supernatant was analyzed.
The blood was diluted to one tenth in PBS, 200 µl was extracted
Rectal swab: 1 ml of DPBS was added, the sample vortexed, then centrifuged 15 min 10000g at room temperature. 800 µl of supernatant was collected for extraction.
We added a comment in the text
iii) the use of the terminology pre-autopsy and at-autopsy is very confusing.
We added the following comment for clarity
Pre-autopsy samples were taken by physicians receiving the infant at the emergency department were from the upper (nasopharynx, throat) and lower respiratory tract (BAL), digestive tract (rectal swab), cerebrospinal fluid (CSF), blood, as well as muscle and heart (sampling done with a trocar)
Samples taken by a pathologist during autopsy (at-autopsy samples) were from the lower respiratory tract (lung), cardiac system (heart), abdominal organs (liver) urinary tract (kidney).
We added a comment in the text
- iv) inexistence of information on the primers used for any of the viruses investigated.
We added a table with the primers for in-house PCRs as supplemental data (Table S1)
Primers for commercial assays are not available
The micrograph panels should present a scale bar.
A scale bar has been added to Figure 1.
Results of the PCR-based analysis showing the values obtained (e.g., viral gene copies) by the different equipment/techniques should be disclosed in the text body or a supplementary data set.
We added a comment in the text
Cycle threshold (Ct) was used as an inverse proxy for viral load,
with lower Ct correlating with higher viral load. In the nasopharygeal wash, a Ct value of 16 was found, that suggest an high viral load.
Using R-gene assays (Biomerieux), in the BAL, despite de facto strong dilution, a Ct value of 31 was found and in the nasal secretions, that are also diluted because of viscosity, a Ct of 25 was obtained that indicate an high viral load Blood was RV-negative.
Film array assays are melting-curve-based multiplex real-time PCR assay for the simultaneous detection of pathogens, there are no Ct or quantitative data.
End point PCR used for rhinovirus typing in nasopharygeal wash, CSF and rectal swab are not quantitative assays.
Reviewer 3 Report
Comments and Suggestions for Authors
Single case report that describes a possible association between rhinovirus infection and SIDS
The clinical case (leading up to the SIDS, such as resuscitation) is described rather short, most of the description is structured around the technical workup. the virological workup is described in a fairly chaotic manner.
CRP is twice 'negative' (line 77 and 78).
Was any analysis done on the CNS parenchyma (signs of encephalitis)?
Was HRV analysis performed on the blood as well (since LP may have been traumatic and HPV detection there may have been contamination from the blood)?
As described in the discussion, HRV has been linked tot SIDS before, so this case is not innovative, nor does it describe a clear cut reason for the pathogenesis of the SIDS (except the changes on lung histology).
Comments on the Quality of English Language
Several typographical and grammatical errors.
Author Response
Single case report that describes a possible association between rhinovirus infection and SIDS
The clinical case (leading up to the SIDS, such as resuscitation) is described rather short, most of the description is structured around the technical workup. the virological workup is described in a fairly chaotic manner.
The day before death, the infant was taken to see physician for congestion. The next day, the infant was found cold and white in his bed, in the supine position. Emergency services were called. On site, the infant was intubated; Gamma adrenalin was administered, the infant was in asystole. He died and was then taken to hospital.
The virological workup description has been improved
CRP is twice 'negative' (line 77 and 78).
It has been corrected
Was any analysis done on the CNS parenchyma (signs of encephalitis)?
There was no histologic analysis of the CNS parenchyma. The whole-body scanner did not reveal encephalitis. Areas of frosted glass and condensation were distributed over both lungs that could be due to infection or post-mortem modifications
We added a comment in the text
Was HRV analysis performed on the blood as well (since LP may have been traumatic and HPV detection there may have been contamination from the blood)?
The CSF was not bloody. As it was slightly hemorrhagic, it was centrifuged before viral analysis, CSF was clear.
To exclude contamination of CSF by blood, we performed a PCR using Rhino&EV/Cc r-gene® kit (Biomérieux), rhinovirus genome was not detected. Blood was also spiked with a HRV positive sample to detect potential inhibitors that could lead to false-negative result. The Ct obtained for the known RV-positive sample and for the known RV-positive sample spiked in the infant blood were similar.
Moreover the Ct obtained for cellular control in the blood and for cellular control in the blood spiked with the HRV positive sample were equal.
The CSF was found RV-positive using two different PCRs, Film array and end-point PCR for typing and sequencing
As described in the discussion, HRV has been linked tot SIDS before, so this case is not innovative, nor does it describe a clear cut reason for the pathogenesis of the SIDS (except the changes on lung histology).
Indeed, there are a few previous published cases from 1970, 1983 and 1998. In this study, we searched for a large range of pathogens using sensitive real–time PCR assays that were able to exclude coinfection, underlining the role of the rhinovirus in SIDS.
The reason for the pathogenesis of the SIDS is multifactorial. I discussed about the infection, the age and sex of the infant, the exposition to smoke, all these items have a role. Regarding the rhinovirus infection, I proposed an explanation: RVs infection in neonates was shown to present an important risk factor of apnea 33. Here, an alveolitis and signs of acute hypoxia have been observed. A dysregulated immune response to rhinovirus infection, with excessive inflammation and damage to the epithelial cells may have resulted in immunopathology and impaired gas exchange in the infant, leading to death.
Round 2
Reviewer 2 Report
Comments and Suggestions for Authors
The Manuscript has been considerably improved.
Minor issues:
Many problems exist with the Primer List present in the supplementary material.
The correct use of RV as an exclusive abbreviation for Rhinovirus must be revised.
L181: please revise punctuation.
Please revise the line spacing throughout the text.
Author Response
Minor issues:
-Many problems exist with the Primer List present in the supplementary material.
The primer list has been improved
-The correct use of RV as an exclusive abbreviation for Rhinovirus must be revised.
It has been revised (L121, L123 and in table 2)
-L181: please revise punctuation.
Done
-Please revise the line spacing throughout the text
Done.